https://doi.org/10.1038/s41467-020-17050-6　　**OPEN**

# Unlocking the passivation nature of the cathode–air interfacial reactions in lithium ion batteries

Lianfeng Zou[1,5], Yang He[1,5], Zhenyu Liu[2,5], Haiping Jia[3], Jian Zhu[4], Jianming Zheng[3], Guofeng Wang[2], Xiaolin Li [3], Jie Xiao [3], Jun Liu [3], Ji-Guang Zhang [3], Guoying Chen [4] & Chongmin Wang [1✉]

It is classically well perceived that cathode–air interfacial reactions, often instantaneous and thermodynamic non-equilibrium, will lead to the formation of interfacial layers, which subsequently, often vitally, control the behaviour and performance of batteries. However, understanding of the nature of cathode–air interfacial reactions remain elusive. Here, using atomic-resolution, time-resolved in-situ environmental transmission electron microscopy and atomistic simulation, we reveal that the cathode–water interfacial reactions can lead to the surface passivation, where the resultant conformal LiOH layers present a critical thickness beyond which the otherwise sustained interfacial reactions are arrested. We rationalize that the passivation behavior is dictated by the Li$^+$-water interaction driven Li-ion de-intercalation, rather than a direct cathode–gas chemical reaction. Further, we show that a thin disordered rocksalt layer formed on the cathode surface can effectively mitigate the surface degradation by suppressing chemical delithiation. The established passivation paradigm opens new venues for the development of novel high-energy and high-stability cathodes.

[1] Environmental Molecular Sciences Laboratory, Pacific Northwest National Laboratory, Richland, WA 99354, USA. [2] Department of Mechanical Engineering and Materials Science, University of Pittsburgh, Pittsburgh, PA 15261, USA. [3] Energy and Environmental Directorate, Pacific Northwest National Laboratory, Richland, WA 99354, USA. [4] Energy Storage and Distributed Resources Division, Lawrence Berkeley National Laboratory, Berkeley, CA 94720, USA. [5]These authors contributed equally: Lianfeng Zou, Yang He, Zhenyu Liu. ✉email: Chongmin.wang@pnnl.gov

nterfacial reactions between a base material and environments are ubiquitous and can deteriorate the performance of a rich variety of materials, including metals, semiconductors, and batteries. Under controlled conditions, however, some of the undesired reactions may become beneficial by assisting the surface or interface passivation, a process that leads to the formation of a stable shielding layer, as well-exemplified by the metal oxides for stainless steel and solid electrolyte interphase (SEI) layers in rechargeable batteries[1–6]. It is commonly observed that the state-of-the-art cathodes for rechargeable batteries, e.g., the nickel-rich and lithium-rich transition metal oxides, are susceptible to, during the stage of fabrication and storage, environmental degradations upon air exposure, a serious issue known as "air instability". Often, electrode instability stems from the cathode–air interfacial reactions by which the resultant products cause many practical issues such as cell degassing, slurry alkalization, electrolyte consumption, and irreversible electrode phase transition[7–14]. Interfacial reactions that leads to surface passivation may offer an effective means for solving the air instability issues of cathode because it can lower the rate of interfacial reactions, and more critically, turn the detrimental interfacial reactions into the beneficial ones.

Unfortunately, the essential elements for establishing the mechanistic understanding of the cathode–air passivation are largely missing. The prevalent understanding of cathode–air instability is obtained based on the techniques such as the X-ray photoelectron spectroscopy (XPS), Thermogravimetric analysis (TGA), and Infrared spectroscopy (IR), which provide the ensemble-averaged information and paint a clear picture of surface composition, including LiOH and $Li_2CO_3$ or a mixture of both[8,9,15–17]. However, such post-mortem characterizations fail to capture the kinetic evolution of the local interfaces required for defining passivation. Further, the cathode–air interfacial reactions are complicated by the composition of air, consisting of multiple components that likely lead to several concurrent reaction events. The lack of one-to-one correspondence between each single reaction and the overall reaction products masks the nature of the interfacial reactions and fails to screen the species that can effectively slow the side reactions, which leads to some simple yet vague mechanisms behind the interfacial reactions. For example, it is believed that the major event occurring at the interface is a direct reaction between the $CO_2$ and the surface residue or the cathodes[9,14,18–20]. In general, besides the composition, establishing the passivation principles for the cathodes requires more characteristic information: morphology of the resultant products such as integrity, uniformity, and crystallographic state; the kinetic information related to the initial stage of interfacial reactions to capture the effects of the reaction products on the interfacial reaction kinetics. It is likely due to the lack of the critical information of the cathode–air interfacial reactions, the passivation design has not translated into the battery field for the prevention of air instability. Environmental transmission electron microscopy (ETEM) offers the route to bridge the gap of the initial stage of cathode oxidation: the high vacuum chamber allows to isolate a single gas for reactions, which is facile to unambiguously identify the kinetic protection of each reaction; the high spatial and temporal resolution enables monitoring the nucleation and subsequent growth kinetics, which is critical for capturing the morphology of surface layers at atomic scale.

In this work, we use atomic level ETEM to in situ probe the dynamics of cathode–air interfacial reactions and establish the fundamental principles for the passivation of cathode in air. We choose the widely commercialized lithium-nickel-magnesium-cobalt based oxides (NMC), including $LiNi_{1/3}Mn_{1/3}Co_{1/3}O_2$(NMC333), $LiNi_{0.6}Mn_{0.2}Co_{0.2}O_2$(NMC622), and $LiNi_{0.8}Mn_{0.1}Co_{0.1}O_2$ (NMC811), as a model system for systematic comparison because air-instability varies

with Ni content[21–26]. By monitoring the interfacial reactions between the cathode and each isolated air component in real-time, we reveal that the cathode–air interfacial reaction is controlled by the $Li^+$-water interactions driven delithiation, rather than chemical reaction. Characteristically, associated with the retarded Li ions diffusion in the reaction products/surface reconstruction layers, the reaction layer shows a critical thickness beyond which the delithiation is arrested, thereby unlocking the potential of self-passivating for the cathode–air interfacial reactions.

## Results

**Chemical delithiation under each air component adsorption.** The surface reactivity of NMC811 in response to each isolated component of air, $N_2$, $O_2$, $CO_2$, and water vapor, is captured at a gas pressure of $5 \times 10^{-2}$ Torr and at room-temperature, as shown in the cross-sectional high-resolution TEM (HRTEM) images of Fig. 1. As expected, the layered structure, seen along [010] zone axis, is chemically resistant to $N_2$ and $O_2$ (Fig. 1a, b), where the NMC811 surface is preserved without forming additional phase upon the continuous gas exposure (lower magnification view is shown in Supplementary Figs. 1 and 2). This is consistent with the post-mortem characterizations that no $Li_2O$ or $Li_3N$ are detected after prolonged exposure in air. Remarkably, despite the numerous documentations for the dressing of $Li_2CO_3$ upon air exposure, the NMC811 is found to be noble against $CO_2$ uptake. The HRTEM image of Fig. 1c, taken after $CO_2$ exposure of 30 mins, displays an intact surface comparable to the one before the exposure, showing no formation of $Li_2CO_3$ (more details in Supplementary Fig. 3). In marked contrast to the case of $N_2$, $O_2$, and $CO_2$, exposing NMC811 to water vapor leads to the formation of a surface layer, with a characteristic thickness of several atomic layers as shown in Fig. 1d, demonstrating that water vapor readily activates the interfacial reactions. The absence of visible contrast in the high angle annular dark field-scanning TEM (HAADF-STEM) while appearance in the annular bright field (ABF)-STEM image for the surface layer suggests that the reaction products are composed of light elements without transition metal components (Fig. 1e, f). High resolution TEM imaging and simulation reveal the surface layers to be LiOH. The LiOH formed on the NMC811 possesses the orientation relationship of LiOH(001)//NMC(001) and LiOH[230]//NMC[1-10], as illustrated in the schematic model of Fig. 1g and the corresponding calculated images of Fig. 1h (more details in Supplementary Fig. 4 and Supplementary Note 1). It may be argued that due to the air exposure of NMC, the surface may be already covered with a native layer of LiOH or $Li_2CO_3$, which can impose significant effects on the Li ions diffusion and affect the in situ ETEM observation of the passivation behavior. However, HRTEM observation of the pristine sample indicates that the native LiOH or $Li_2CO_3$ is not continuous, showing features of clean surface segment and therefore lending the convenience for in situ ETEM observation of the passivation behavior (Supplementary Fig. 5). Apparently, if the thickness of Li related compound is below the critical thickness, the delithiation will continues until reaching a critical thickness; if the thickness of Li compound islands is larger than the critical thickness, the Li compound will remain inert.

Early notions considered the air instability of cathode is caused by the direct reactions between cathode and gases. Given the high reactivity between Li ions and the air species, the reaction products, typically LiOH and $Li_2CO_3$, are expected to form on NMC surfaces. However, this reaction-dominant mechanism essentially fails to yield behavior consistent with the real time observations by which the cathode-gas reaction products are only noticed under water vapor exposure, while being absent in $O_2$, $N_2$, and $CO_2$. This discrepancy lies in the reaction-dominant

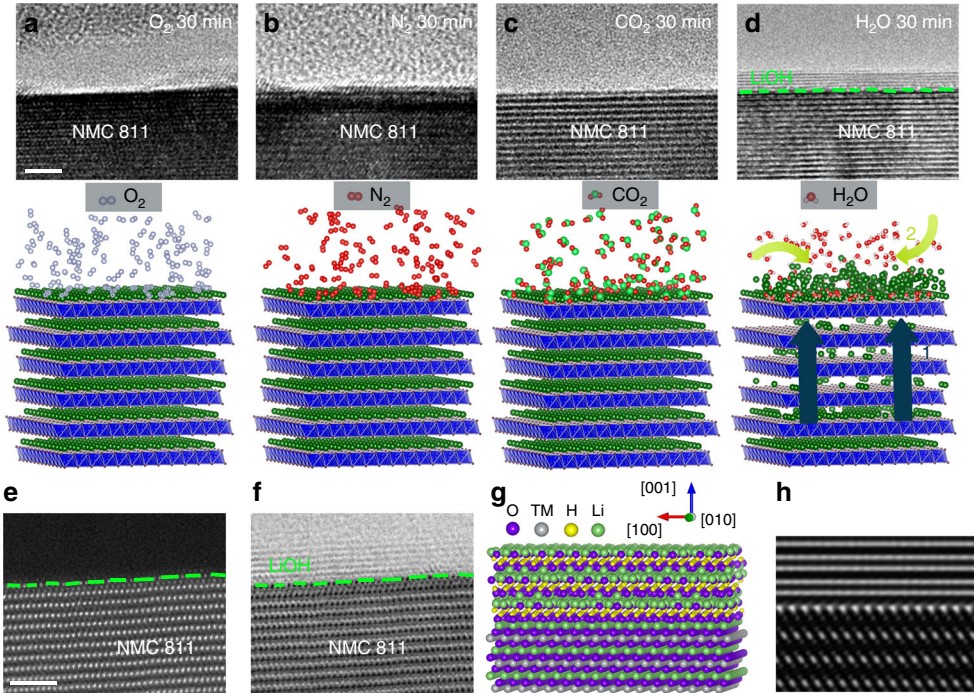

**Fig. 1 The chemical delithiation of NMC 811 in air.** The green dashed lines outline the surface boundaries of NMC811. **a–d** The distinct capability of each air component in delithiating the NMC811. The upper panels are the HRTEM images of NMC 811 surface after exposure to $N_2$, $O_2$, $CO_2$, and water vapor at the gas pressure of $5 \times 10^{-2}$ Torr and at the room temperature for 30 mins, respectively. The lower panels show the schematic drawing of Li ions evolution in the layered structure under the exposure of each corresponding gas. The label 1 and 2 in **d** indicate the typical two steps for the formation of LiOH: the Li diffusion from bulk to surface, followed by the surface reactions. **e, f** Identification of surface reaction products by HAADF-STEM and the corresponding ABF-STEM imaging of NMC811 after prolonged exposure in water vapor. **g** Atomistic model of LiOH on NMC substrate with the alignment of LiOH(001)// NMC(001) and LiOH[230]//NMC[1-10]. **h** The simulated HRTEM image based on the model in **g**. Scale bar, 2 nm **a, e**.

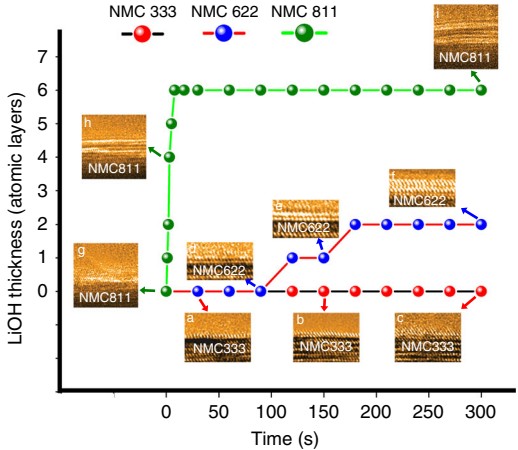

**Fig. 2 Growth kinetics of interfacial reaction layers.** The in situ measured LiOH passivation layer thickness as a function of exposure time for **a–c** NMC333, **d–f** NMC622, and **g–i** NMC811 at $P_{H2O} = 5 \times 10^{-2}$ Torr and room temperature. The growth of LiOH passivation layer stops a certain critical thickness for each NMC compound, and the critical thickness scales with the Ni concentration. The inserts are the HRTEM images extracted from each time-lapsed sequence. The arrows point to the inserts corresponding to each time point during the time sequences.

mechanism is only valid when the reactants, in this particular scenario, the Li ions and gases, are readily available on surfaces for reactions. However, the supply of Li ions in the case of cathode requires the diffusion of Li ions from bulk lattice to the surfaces, which can be largely controlled by gas-Li ions interactions. In light of this, the delithiation driven by the Li

ions-gas interactions is the critical step that governs the cathode-gas reactions and the reaction layer thickness. The present in situ observations indicate that the water vapor severs as the only active component in air that triggers the delithiation process.

**Passivation of the LiOH film at the cathode–air interface.** Once delithiation process is activated, the continuous water vapor uptake is expected to continuously delithiate NMC to lead to a persistent cathode-water vapor interfacial reaction and thickening of hydroxide layers. However, the formation of hydroxide layer dramatically affects Li extraction kinetics, as demonstrated by hydroxide growth at a constant pressure of $P_{H2O} = 5 \times 10^{-2}$ Torr and at room temperature. The time-lapsed images in Fig. 2a–c show that the NMC333 surface is inactive against the water vapor. Despite the appearance of precursor-like species, the hydroxide layer barely develops into a monolayer coverage throughout the whole water vapor exposure sequences (Supplementary Fig. 6 and Supplementary Movie 1), indicating the intrinsic high barriers against chemical delithiation for NMC with Ni-less composition. By contrast, the NMC622 surfaces are noticed to be readily passivated by an ultrathin hydroxide layer. After the water vapor exposure of ~125 s, a crystalline layer is seen to spread in two dimensions and epitaxially oriented with NMC622 (Fig. 2d, e). With a short period of incubation, a second layer completely covers the surface with a similar lateral propagation (Fig. 2f). Critically, the thickness of the hydroxide layer remains constant even with a subsequent prolonged water vapor exposure, presenting a typical feature of self-limiting growth (Supplementary Fig. 7 and Supplementary Movie 2). Under the identical conditions, the nucleation and growth of LiOH layers on NMC811 is much faster than that on NMC622, without the presence of incubation period. A single LiOH layer emerges on

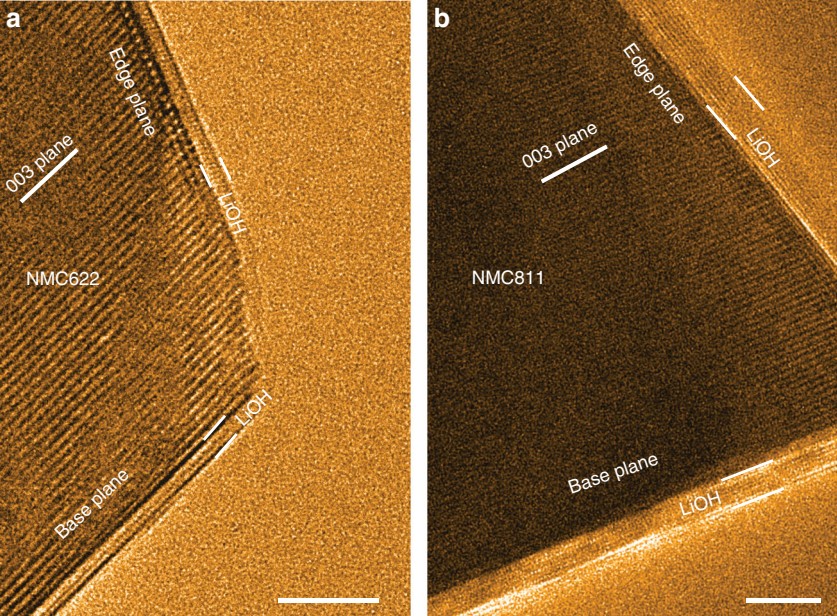

**Fig. 3 Passivation layers on both the edge-plane and base-plane in NMC.** The images are taken after exposing samples to $H_2O$ vapor ($PH_2O = 5 \times 10^{-2}$ Torr) for 30 mins. **a** Surface of NMC622. **b** Surface of NMC811. Scale bar, 5 nm **a**, **b**.

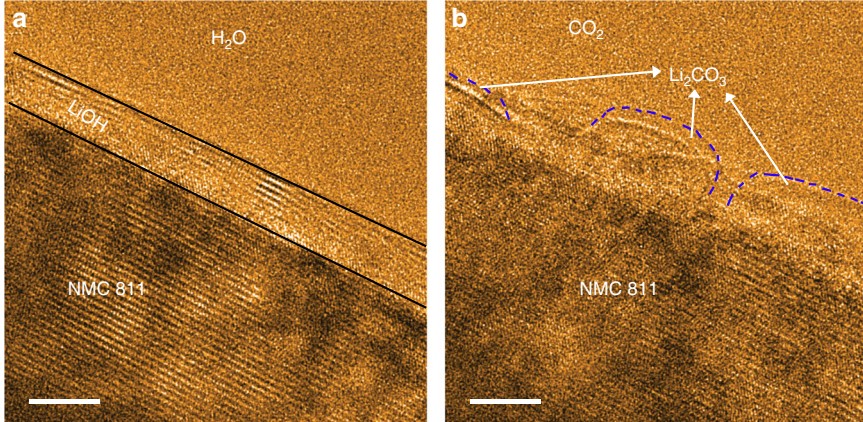

**Fig. 4 Passivation layers evolution with the presence of $CO_2$.** **a** The surface of NMC811 after exposed to water vapor ($PH_2O = 5 \times 10^{-2}$ Torr) for 30 mins, indicating a uniform passivation layer of LiOH. The black lines outline the boundary of LiOH. **b** The surface morphology of NMC811 after the water vapor exposure and followed by exposure to $CO_2$ for 10 mins ($PCO_2 = 5 \times 10^{-2}$ Torr), leading to the formation of $Li_2CO_3$. The blue dashed lines outline the boundary of each $Li_2CO_3$ island. Scale bar, 5 nm **a**, **b**.

the particle surface immediately after the water vapor uptake (Fig. 2g) and the LiOH layer continuously thickens (Fig. 2h) to reach a saturated plateau at 20 s (Fig. 2i). Similar to NMC622, further water vapor exposure does not lead to the continued growth of LiOH, rather it stays at a critical thickness of ~6 atomic layers in the subsequent ~25 mins (Supplementary Fig. 8 and Supplementary Movie 3), further confirming the characteristics of self-limiting growth. These observations clearly demonstrate that the cathode–air interfacial reaction corresponds to a passivation process with the limiting step being the delithiation of cathode and featuring a critical thickness of passivation layer.

**Integrity of the passivation films**. The global features of passivating layers, such as the continuity and homogeneity, are captured by the beam blank experiment. A basal plane of pristine NMC811 is targeted and a low magnification is selected for the observations, as shown in Supplementary Fig. 9a. After

exposure of 30 mins in water vapor, with electron beam blank and at a constant water vapor pressure of $5 \times 10^{-2}$ Torr, a conformal passivation layer appears on the entire particles surface with a uniform thickness comparable to that in Figs. 1d and 3b, and Supplementary Fig. 9b. This observation also demonstrates that the electron beam dosage used for in situ observations has negligible influence on the passivation layer growth. Meanwhile, our in situ ETEM observation of primary particle (Fig. 3a) displays the same critical thickness of passivation layer as that of the secondary particle prepared by focused ion beam (FIB), suggesting the limited amount of impurities introduced by the FIB sample preparation barely alters the surface passivation behavior and again, rationalizes that the interfacial passivation behavior is intrinsically self-limiting.

Interestingly, although the $Li^+$ bears distinct diffusion barriers in the direction along-channel and across-channel for the layer structured NMC, our in situ ETEM results demonstrate that both the base-plane and edge-plane of NMC622 and NMC811 are

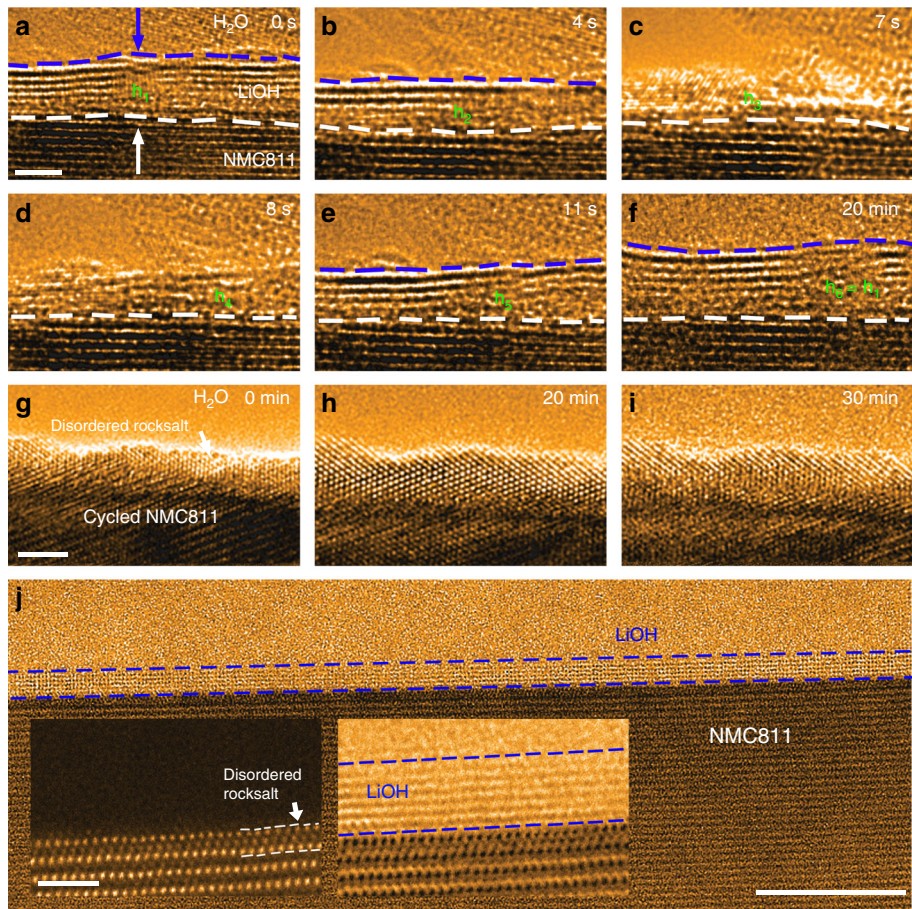

**Fig. 5 Self-healing and disordered rocksalt resultant passivation.** All the TEM sequences are captured under the continuous exposure to water vapor with a pressure of $P_{H2O} = 5 \times 10^{-2}$ Torr and at room temperature. **a** Pristine NMC811 covered with a saturated hydroxide film after the water vapor exposure of 30 mins with the electron beam blanked. **b**, **c** The hydroxide thickness evolution with the condense of the electron beam. The saturated LiOH layer with a thickness of h1 is completely removed after 7 s. **d**–**f** The self-healing processes at the local breaches as the weakening of electron beam. The LiOH layer recovers to the characteristic thickness after a few seconds and maintains the thickness during the subsequent 20 mins. **g**–**i** The surface evolution of NMC811 with a disordered rock salt shell. The arrow points to the disordered phase developed upon cycling, no reaction products are formed during the 30 mins water vapor exposure. **j** ABF-STEM images of the NMC811 after being exposed to water vapor for 30 mins. The contour of LiOH is outlined by the blue dashed lines. The inserts in **j** are the HAADF-STEM (left) and the corresponding ABF-STEM image (right) showing the surface with half-unit-cell-thick disordered structure and the formation of saturated LiOH layers after the water vapor exposure. The white and blue dashed lines indicate the boundaries of disordered rocksalt structure and the LiOH layers. Scale bar, 3 nm **a**, 2 nm **g**, 10 nm **j**, 1 nm inserts of **j**.

passivated by the LiOH layers with the same critical thickness as shown in Fig. 3, indicating that the Li–H₂O interaction provides a strong driving force for the Li deintercalation and is the determining factor for the passivation layer thickness. However, the presence of $CO_2$ introduces a permanent damage to the LiOH passivation layers by which the carbonation reactions on surfaces result in the evolution from a flat conformal LiOH passivation layer to discrete islands of $Li_2CO_3$, leaving gaps that allow the $Li^+$ deintercalations and thus sustained reactions (Fig. 4a, b). The broken of the integrity of the LiOH layer with the presence of $CO_2$ also explains why the Li ion insulator – $Li_2CO_3$ – is not playing an effective role in impeding the sustained cathode–air reactions.

**Self-healing of LiOH passivation films.** To investigate the dynamic evolution of passivation films in response to the potential physical damages and further rationalize the atomistic origin and passivation nature of cathode–air interfacial reactions, a delicate experiment is designed. The electron beam is utilized to mimic the external stimuli as the passivation layers are fragile against the high beam dosage. A segment of passivated surface

capped with the saturated hydroxide is randomly selected and zoomed in (Fig. 5a), and the electron beam is deliberately condensed to generate the local damages. The native passivation layers start to decompose and become thinner under the high beam current (Fig. 5b) and disappear after 7 s (Fig. 5c). Intriguingly, with the lowering of electron beam dosage, the damaged section of passivation layer begins a "auto-repair" process (or "self-healing") through dynamic delithiation and reactions (Fig. 5d), which progressively patch the bared surface (Fig. 5e) and eventually recover to the characteristic thickness (Fig. 5f, Supplementary Fig. 10 and Supplementary Movie 4). Such prompt auto-repairs of local damages are pivotal for retaining the passivation layer integrity, more importantly, with the abundant Li ions supply from NMC reservoir, the self-healing of local breakdowns may represent regular events in the context of moisture exposure due to the inherently delithiating capability of water vapor.

The observations of self-healing behavior also provide a wealth of underpinnings on the nature of chemical passivation. The growth of hydroxide ceases at the critical thickness while the formation of local breakdown reactivates the growth delineates

the Li ions transport paths for interfacial reactions: starting from NMC parent phase and percolating through the grown films. This simultaneously identifies the Li ion sources that fed the hydroxide growth are originated from the NMC bulk rather than the excess Li species on surfaces (Supplementary Fig. 11 and Supplementary Note 2). Moreover, the auto-repairing of hydroxide after damaging the saturated thin films concludes that the Li ions in the bulk lattice of NMC are sufficient for further reactions, which rules out that the self-limited growth is caused by the draining of Li ions.

**Disordered rocksalt suppresses chemical delithiation**. Formation and thickening of cathode–air interface layers are controlled by the delithiation, therefore, suppressing the chemical delithiation can lead to the control of the interfacial reactions. Given the general paradigm that the ordered structure favors the Li ions transport while the disordered structure limits[27–29], a shell with the disordered structure on cathode is envisaged to be chemically resistive and effectively protects the surface from chemical delithiation. Here we produce the surface disordered structure by utilizing the inherent layered-to-disordered rocksalt phase transition associated with electrochemical cycling in NMC and rationalize the beneficial effects of disordered surface structure on the prevention of chemical delithiation. A disordered rocksalt shell, with a thickness of ~2 nm, is poised to cover the primary particle after 10 cycles (Fig. 5g). Indeed, the subsequent exposure of the cycled NMC811 in water vapor barely leads to any surface reaction products by the end of the sequence (Fig. 5h, i, Supplementary Fig. 12 and Supplementary Movie 5). This observation indicates that the 2-nm-thick rocksalt shell provides sufficient barrier against chemical delithiation and the subsequent formation of surface LiOH layer. However, the half-unit-cell-thick disordered rocksalt fails to impede the Li extraction (Fig. 5j), projecting that the Li de-intercalation barriers scales with the thickness of disordered structure, and a certain thickness is required to effectively hinder the chemically driven Li depletion. For the studies of the surface rocksalt structure on the surface passivation, following the FIB lift out of the sample, the sample is checked carefully with TEM imaging to ensure the rocksalt surface without pre-existing CEI layers to carry out the in situ ETEM studies (Fig. 5g).

## Discussion

During the electrochemical cycling, once a stable SEI layer is grown, the rate of electrode–electrolyte interfacial side reactions slows dramatically due to the sluggish electrons transport across the passivation layer, thereby mitigating many detrimental effects such as the consumption of Li ions[30,31], phase transitions[32–35], intergranular cracks[36–39], and impedance rise[40]. By contrast, our results indicate that the origin and the dynamics of cathode–air interfacial reactions are controlled by the mass transport, that is, the Li ions percolation from bulk towards surface. In a reactive environment, the compositional evolution of a solute-host system is often driven by the solute-gas interactions by which the stronger affinity results in a higher propensity for the ionic specie to come to the outer surface. Clearly, our experimental observations indicate that the water vapor is the sole specie that provides sufficient driving force for the Li ions depletion. To substantiate the critical roles of water vapor in determining the cathode–air interface related behavior, the Li segregation energies, $\Delta E$, in each individual gas were calculated and shown in Fig. 6c. Here the $\Delta E$ is defined by the energy difference by moving a Li ion from the subsurface to surface 3b site (Fig. 6a, b and Supplementary Fig. 13), therefore, a negative $\Delta E$ indicates that the Li ions are attracted by gases and tend to reside on surfaces; on the contrary,

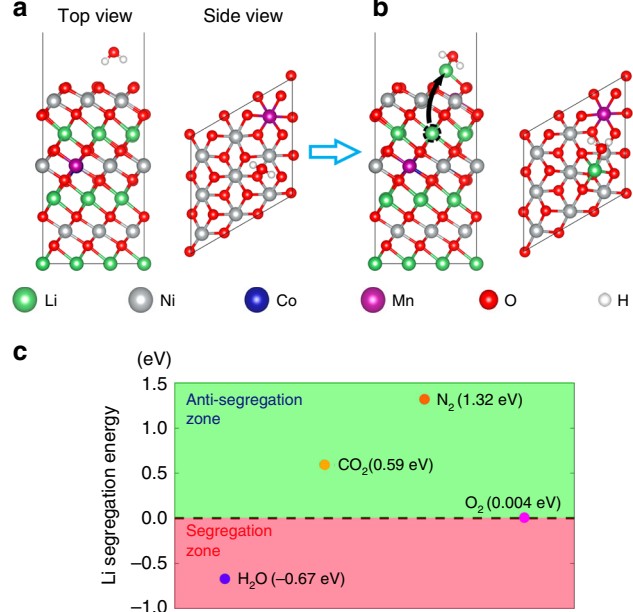

**Fig. 6 Li segregation energies under the adsorption of air components. a, b** The atomic structures of conventional NMC, and delithiated NMC with a Li ion moves from the subsurface to the surface 3b sites under gas molecule adsorption. The dashed black ring indicates the migrated Li ions and the black arrow points to the surface site that the sub-surface Li migrates to. **c** The plot of Li segregation energies under the water vapor, $O_2$, $CO_2$, and $N_2$ adsorption. The upper and lower regime represent the Li anti-segregation, and segregation zone, respectively.

a positive $\Delta E$ suggests that Li$^+$ segregation on surface is unfavorable. Consistent with our TEM observations, the $\Delta E$ shows a highly negative value as the adsorption of water vapor, indicating the strong propensity of Li ions to leave the host structure for surface passivation. As a comparison, the segregation energies reverse as the uptake of the alternative gases in air and follow the repulsion order of $\Delta E_{O2} < \Delta E_{CO2} < \Delta E_{NO2}$ (Fig. 6c), which all fall into the anti-segregation zone, featured by the occupation of Li ions in the bulk.

Using layered cathode as a model system, we reveal the origin and the nature of cathode passivation in air. Instead of the straightforward chemical reactions, the heart of cathode–air interfacial reactions lies in the deilithiation that is essentially driven by the interactions between Li ions and water vapor species, which consequently governs the self-limiting growth and the self-healing processes of the passivation layers at interfaces. The observations that interfacial reaction kinetics can be modified by structuring of cathode surface provide pathways for the passivation design and hold technological promises for mitigating the detrimental consequences associated with surface reaction. We expect the precise engineering of cathode surface structure can lead to sluggish cathode–air interfacial reactions, paving the way for electrodes protection through actively controlling of chemical delithiation processes.

## Methods
**Preparation of the NMC electrode and disordered rock salt shell**. The secondary particles are purchased from Targrey, and the primary particles are synthesized at Lawrence Berkeley National Laboratory. The NMC333, NMC622, and NMC811 pristine electrodes were prepared by dissolving the mixture of NMC powder, Super P, and polyvinylidene difluoride with a ratio of 80: 10: 10 in N-methyl-2-pyrrolidone. The layered-disordered rock salt core–shell structure was produced by electrochemical cycling of NMC811, using CR2032-type coin cells with Celgard 2500 as separator, high-purity metallic lithium as counter electrode, and 1 M LiPF6 in ethylene carbonate and ethyl methyl carbonate (4:6, w/w) as

electrolyte. The electrochemical cycling was performed at the temperature of 30 °C and at the rate of 0.1C. The cut-off voltage was set to be 4.3 V vs. Li/Li+ for the charge process and 2.7 V vs. Li/Li+ for the discharge process. The NMC811 cathode was harvested from the disassembled cells after 10 cycles and soaked in dimethyl carbonate (DMC) for 24 h before TEM experiments.

**TEM sample preparation and characterizations**. The as-prepared electrodes consisting of secondary particles were transferred to the FEI Helios Dual Beam system with an operation voltage ranges from 2 to 30 kV was employed to carry out the lift out. Several random secondary particles of the electrode were targeted and the lift out processes were started by coating the targeted particles with Pt layers of ~2 μm thickness. After that, a ~3-μm cube was extracted out of the secondary particle using focused ion beam (FIB), followed by attaching on the Cu TEM grid. The subsequent thinning processes was performed step by step using a voltage of 30 kV until reach electron transparent thickness, the final polishing processes were carried out at a lower voltage of 5 kV and 2 kV.

The as-prepared samples are then transferred to ETEM for the gas experiments. Before each gas is introduced for reactions and in situ observations, the TEM column was pumped to $10^{-8}$ Torr and then a single gas was introduced into the column each time for the gas reactions. The beam current used for imaging is ~2 nA. The STEM images were performed on the aberration corrected JEOL JEM-ARM200CF, at the operation voltage 200 kV. The electrons range from 90 mrad to 370 mrad, and 10 mrad to 23 mrad were collected for HAADF-STEM, and ABF-STEM imaging, respectively.

**Atomistic simulation method**. The first principles density functional theory (DFT) calculations has been performed to investigate the Li stability of NMC under different gas adsorptions. The surface of the NMC was modeled by a slab structure with 27 transition metal (TM) atoms (21 Ni atoms, 3 Mn atoms, and 3 Co atoms) in the simulation cell, as shown in Fig. 6 and Supplementary Fig. 13. During structural relaxation, the positions of the bottom four layer of atoms were fixed. Each of the $H_2O$, $CO_2$, $N_2$, and $O_2$ molecule was adsorbed on the NMC surface. Subsequently, one Li atom at subsurface layer was allowed to migrate to the surface 3b site, leaving a Li vacancy in the subsurface layer. The energy difference of the adsorption configuration before and after the Li atom migration (Fig. 6 and Supplementary Fig. 13) was then calculated to evaluate the tendency of Li to come to NMC surface.

All the DFT calculations were performed using the Vienna Ab-initio Simulation Package (VASP), with plane wave basis and Projector Augmented Wave (PAW) formulism[41,42]. The generalized gradient approximation (GGA) in the form of Perdew, Burke and Ernzernhof (PBE)[43] functionals was used to evaluate the electronic exchange and correlation energy. For Ni, Co, and Mn, the Hubbard-U correction was employed with the effective on-site interaction parameter Ueff to be 6.4 eV, 4.9 eV, and 4.5 eV as suggested in refs. [44–46]. A kinetic energy cutoff of 500 eV was used for plane wave expansion. For the electronic relaxation, the total energy was converged to $10^{-5}$ eV. For the ionic relaxation, the force acting on each atom was converged to 0.01 eV/Å. A Γ centered k-point mesh of $4 \times 4 \times 1$ was used to sample the Brillouin zone.

## Data availability
All data that support the findings of this study have been included in the main text, Supplementary information, and Supplementary movies. Original data are kept at the Environmental Molecular Sciences Laboratory at Pacific Northwest National Laboratory and are available from the corresponding authors upon reasonable request.

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

## Acknowledgements

This work is supported by the Assistant Secretary for Energy Efficiency and Renewable Energy, Office of Vehicle Technologies of the U. S. Department of Energy under the advanced cathode materials program (award DE-LC-000L053). The work was conducted in the William R. Wiley Environmental Molecular Sciences Laboratory (EMSL), a national scientific user facility sponsored by DOE's Office of Biological and Environmental Research and located at PNNL. PNNL is operated by Battelle for the Department of Energy under Contract DE-AC05-76RLO1830. We gratefully acknowledge the computational resources provided by the computer facility at Center for Simulation and Modeling of the University of Pittsburgh and at the Extreme Science and Engineering Discovery Environment (XSEDE), which is supported by National Science Foundation grant number ACI-1053575.

## Author contributions

L.Z. and C.W. conceived the research plan, Y.H. and L.Z. prepared the TEM samples and carried out the in situ ETEM study, H.J., J.Z., J.M.Z., X.L., J.X., J.L., J.G.Z., and G.C. prepared the electrode materials and performed the electrochemical cycling, Z.L. and G. W. carried out the computational work. L.Z and C.W. analyzed the data and drafted the manuscript with final approval by all authors.

## Competing interests

The authors declare no competing interests.
