## [Peer Review File · Nature Communications]

Reviewer #1 (Remarks to the Author):

In this manuscript, the authors reported on the chemical delithiation that results in the air-instability of Ni-rich cathodes, and used the LiOH and disordered rocksalt structures to rationalize the surface passivation of the NMC cathodes. Many industrial and academic efforts have been made to increase the nickel content of NMC due to its higher capacity relative to Co and the limited stocks of Co, however, one of the remaining issues for commercializing high nickel NMC is the instability upon the air exposure. The use of dynamic, atomic-scale electron microscopy observations realizes the observations of each isolated reaction and thus delineates the origin of the air instability. The authors used the NMC cathodes as a model systems for this study, however, the results can possibly provide a broader guideline for tackling the air related issues of many advanced cathodes such as lithium rich cathodes because the fundamentals of chemical-driven delithiation may have a general applicability, so it satisfies the requirements to be published in a high-impact journal. The texts and figures were organized in a logical way, I recommend acceptance of the manuscript after some minor revisions.

1.The current data presents a comprehensive picture of the passivation processes on the (003) terminated surface, to achieve effective surface protection, the edge planes of NMC need to be covered as well. However, the lithiation diffusion barriers can be different in the along-channel and across-channel directions. Would the edge planes, e.g., the (003) exit surfaces, be passivated as well?

2.The real-time TEM observations suggest that the CO₂ barely contribute to the initial stage of chemical delithiation, however, it would eventually alter the final products in the subsequent reactions with the passivation layers, what are the effects of such reactions on the developed passivation layers? Would this change the passivation nature of surface films?

3.The TEM lamella used for the in situ experiments is prepared by FIB. Despite the final cleaning process performed at 2kV, a thin surface damage layer would be remaining there and the trace Ga could be embedded in the samples, how would the impurities affect the formation of the surface passivation layers?

4.The details for the TEM experiments are missing.

5.The format of references has many issues, please check and fix them carefully.

Reviewer #2 (Remarks to the Author):

The manuscript entitled "Unlocking the nature of the cathode-air interfacial reaction in lithium ion batteries" reports the in situ TEM observation of cathode surfaces exposed to various gas atmospheres. This paper claims that water vapor is the only active species that react with cathode surface forming a LiOH passive layer via chemical delithiation. Although the gas pressure used here is still much lower than the actual condition, the experimental data well corroborate the current understanding of air-instability of Ni-rich NMC cathodes. More interestingly, the authors show that a thin disordered rock-salt layer formed on the NMC811 surface can effectively mitigate the surface degradation by suppressing chemical delithiation. This observation may open a new venue for the development of novel high-energy and high-stability cathodes. Therefore, I recommend publication of this paper after the following issues are addressed properly.

Comments:

(1) The moisture sensitivity of Ni-rich cathodes and the formation mechanism of residual lithium compounds such as LiOH and Li₂CO₃ via reaction of delithiated lithium with water and CO₂ have already been proposed and extensively studied in the literature. I don't see this paper challenges

the current understanding of surface degradation upon air-exposure. It rather corroborates the existing ex situ experiments data. In an actual air-storage situation, moisture in the air would initiate chemical delithiation to form a LiOH layer and subsequently, LiOH reacts with CO₂ forming Li₂CO₃. The main text should be revised to address this.

(2) An as-received pristine Ni-rich NMC particle is not even clean on the surface; LiOH and Li₂CO₃ are present as a residual Li. The content and composition of such residual Li species also have a significant effect on the air-instability of NMCs. How were the NMC samples prepared and treated before the in situ TEM experiment? Please provide more information about the sample sources and the initial surface conditions should be characterized more carefully.

(3) The stabilization of the cycled electrode is attributed to the disordered rock-salt layer. However, the surface of the multiple-cycled cathode would be already protected by a thick SEI layer, which may not be completely cleaned away by rinsing with the DMC solvent.

Reviewer #1 (Remarks to the Author):

Comment: In this manuscript, the authors reported on the chemical delithiation that results in the air-instability of Ni-rich cathodes, and used the LiOH and disordered rocksalt structures to rationalize the surface passivation of the NMC cathodes. Many industrial and academic efforts have been made to increase the nickel content of NMC due to its higher capacity relative to Co and the limited stocks of Co, however, one of the remaining issues for commercializing high nickel NMC is the instability upon the air exposure. The use of dynamic, atomic-scale electron microscopy observations realizes the observations of each isolated reaction and thus delineates the origin of the air instability. The authors used the NMC cathodes as a model systems for this study, however, the results can possibly provide a broader guideline for tackling the air related issues of many advanced cathodes such as lithium rich cathodes because the fundamentals of chemical-driven delithiation may have a general applicability, so it satisfies the requirements to be published in a high-impact journal. The texts and figures were organized in a logical way, I recommend acceptance of the manuscript after some minor revisions.

Reply: Thanks to the review and we appreciate your positive comments and insightful assessments of our results, which we found are very useful for enhancing the readability of the manuscript upon revisions as suggested by you.

Question 1. The current data presents a comprehensive picture of the passivation processes on the (003) terminated surface, to achieve effective surface protection, the edge planes of NMC need to be covered as well. However, the lithiation diffusion barriers can be different in the along-channel and across-channel directions. Would the edge planes, e.g., the (003) exit surfaces, be passivated as well?

Answer to question 1: This is a great point. The Li ions indeed present an anisotropic diffusion behavior in the along-channel and across-channel directions upon electrochemical cycling, which can possibly lead to the distinct reaction kinetics on different facets. Often, the diffusion barrier of Li^+ along the (003) channels is much lower than that of across (003) channel direction, implying the Li^+ can migrate to the edge planes easier than to the (003) terminated surfaces. Given the (003) base planes have developed the continuous passivation layers, it is reasonable to expect the edge plane can also form a passivation layer because of the more ready supply of Li resources.

We have performed additional experiments to verify this hypothesis. A corner area of a particle with the presence of both base plane and edge plane has been selected for the observation. The **added Fig.3** shows the HRTEM image of surface morphology after exposing to H_2O (pressure of $\text{PH}_2\text{O}=5 \times 10^{-2}$ Torr) for 30 min, which clearly demonstrates that a well-developed conformal passivation layers develops on the two facets of both base-plane and edge-plane for the case of both NMC622 and NMC811. Interestingly, it seems that the passivation layers are of same thickness on the base-plane and the edge-plane, indicating that the Li- H_2O interactions provide a strong driving force for the Li deintercalation and is the determining factor for the passivation layer thickness.

To accommodate this point, upon revision, we have incorporated the new HRTEM image as Fig.3. We have also added the relevant discussion as follows:

“Interestingly, although the Li^+ bears distinct diffusion barriers in the direction along-channel and across-channel for the layer structured NMC, our in-situ ETEM results demonstrate that both the base-plane and edge-plane of NMC622 and NMC811 are passivated by the LiOH layers with the same critical thickness as shown in Fig.3, indicating that the Li- H_2O interaction provides a strong driving force for the Li deintercalation and is the determining factor for the passivation layer thickness.”

Added as Fig.3. HRTEM image of passivation layers on the edge-plane and base-plane of the NMC cathodes. The images are taken after exposing the samples to H_2O vapor ($\text{PH}_2\text{O}=5 \times 10^{-2}$ Torr) for 30 mins. The white lines outline the boundary of LiOH layers. (a) Surface of NMC622. (b). Surface of NMC811

Question 2. The real-time TEM observations suggest that the CO₂ barely contribute to the initial stage of chemical delithiation, however, it would eventually alter the final products in the subsequent reactions with the passivation layers, what are the effects of such reactions on the developed passivation layers? Would this change the passivation nature of surface films?

Answer to question 2: That is a great question and we are very appreciative of the comments. As you rightly pointed out, the CO₂ can react with LiOH and possibly destroy the passivation nature of the developed surface layers. To clarify this, per your suggestion, we have performed additional experiments to observe the effects of carbonation reactions on the passivation nature of LiOH layers. The same procedures were used to develop LiOH passivation layers by which the pristine NMC811 were firstly exposed to the water vapor ($P_{H_2O}=5 \times 10^{-2}$ Torr) for 30mins to grow a saturated film, as shown in the added Fig.4a. The H₂O vapor supply was then shut off and the residue H₂O vapor was evacuated, after that, the CO₂ with a pressure of $P_{CO_2}=5 \times 10^{-2}$ Torr was introduced inside the ETEM column to initiate further reactions. During the carbonation reactions, the electron beam was blanked to reduce any potential beam effects. The surface morphology was captured after 10mins exposure in CO₂. As shown in the added Fig.4b, the morphology of the surface layers has transformed from a flat conformal layers of LiOH into the discrete islands of Li₂CO₃, leaving gaps at between the islands. Given the integrity of the surface films is critical for the passivation of NMC cathode, although part of the surface covers with Li₂CO₃, the bared surface arising from the carbonation reactions would provide express paths for the delithiation, which eventually deteriorate the passivation nature.

To accommodate this point, we have added this discussion into the revision as follows:

“However, the presence of CO₂ introduces a permanent damage to the LiOH passivation layers (Fig.4a) by which the carbonation reactions on surfaces results in the evolution from a flat conformal LiOH passivation layer to discrete islands of Li₂CO₃, leaving gaps that allow the Li⁺ deintercalations and thus sustained reactions (Fig.4b). The broken of the integrity of the LiOH layer with the presence of CO₂ also explains why the Li ion insulator of Li₂CO₃ is not playing an effective role in impeding the sustained cathode-air reactions.”

Added as Fig.4. Evolution of the passivation film with the presence of CO₂. (a) the surface morphology of NMC811 after exposed to the water vapor ($P_{H_2O}=5 \times 10^{-2}$ Torr) for 30mins. (b) the surface morphology of NMC811 after exposed to the CO₂ for another 10mins ($P_{CO_2}=5 \times 10^{-2}$ Torr). The blue dashed lines outline the boundary of each Li₂CO₃ island.

Question 3. The TEM lamella used for the in situ experiments is prepared by FIB. Despite the final cleaning process performed at 2kV, a thin surface damage layer would be remaining there and the trace Ga could be embedded in the samples, how would the impurities affect the formation of the surface passivation layers?

Answer to question 3: Thank you for raising this great point. As you rightly pointed out, the impurities in the materials system can possibly affect the ion diffusivity at some local sites. To examine the effects of the impurities and the surface damage layers on the observed behaviors, we use as-prepared NMC622 primary particles to make a comparison with the FIB lifted out secondary particles. Note the primary particles are of highly anisotropic shape with a dimension that is electron transparent, which can be directly loaded on the TEM grid for the ETEM observations, omitting the unnecessary sample preparation procedures and thus avoid the potential surface damage and the Ga impurities that can be introduced by the FIB method. Our results show that the surface passivation layer is of same critical thickness as that of FIBed sampled, as representatively shown in Fig. 3a. Therefore, with the careful cleaning at lower voltages, the surface damage layer and the limited amount of Ga has a negligible effect on the percolation of the Li ions and the subsequent passivation formation on the NMC surfaces.

To address this point, we added the following sentences into the main text: “It should be pointed out that our in-situ ETEM observation of primary particle (Fig.3a) displays the same critical thickness of passivation layer as that of the secondary particle prepared by focused ion beam (FIB), suggesting the limited amount of impurities introduced by the FIB sample preparation

barely alter the surface passivation behavior and again, rationalizes that the interfacial passivation behavior is intrinsically self-limiting.”

Question 4. The details for the TEM experiments are missing.

Answer to Question 4: Thank you for the careful reading. We have incorporated the beam current information for TEM experiments in the revised version.

Question 5. The format of references has many issues, please check and fix them carefully.

Answer to Question 5: Thanks to the reviewer for noticing this point. We have corrected the references to fit for the Nature Communications format.

Reviewer #2 (Remarks to the Author):

Comment: The manuscript entitled “Unlocking the nature of the cathode-air interfacial reaction in lithium ion batteries” reports the in situ TEM observation of cathode surfaces exposed to various gas atmospheres. This paper claims that water vapor is the only active species that react with cathode surface forming a LiOH passive layer via chemical delithiation. Although the gas pressure used here is still much lower than the actual condition, the experimental data well corroborate the current understanding of air-instability of Ni-rich NMC cathodes. More interestingly, the authors show that a thin disordered rock-salt layer formed on the NMC811 surface can effectively mitigate the surface degradation by suppressing chemical delithiation. This observation may open a new venue for the development of novel high-energy and high-stability cathodes. Therefore, I recommend publication of this paper after the following issues are addressed properly.

Reply: Thanks to the reviewer and we greatly appreciate your encouraging comments. We also thank you for taking time and energy to offer us insightful comments and suggestions, which serves to guide us for the revision of the manuscript for clarity.

Question 1: The moisture sensitivity of Ni-rich cathodes and the formation mechanism of residual lithium compounds such as LiOH and Li_2CO_3 via reaction of delithiated lithium with water and CO_2 have already been proposed and extensively studied in the literature. I don't see this paper challenges the current understanding of surface degradation up on air-exposure. It rather corroborates the existing ex situ experiments data. In an actual air-storage situation, moisture in the air would initiate chemical delithiation to form a LiOH layer and subsequently, LiOH reacts with CO_2 forming Li_2CO_3 . The main text should be revised to address this.

Answer to question 1: This is a great point that made us think a great deal about the clarification of the significance. As you rightly pointed out, extended effort has been made on the understanding of the detrimental effects of the air exposure, which successfully correlate the air instability with the formation of LiOH and Li_2CO_3 on cathode surface. The classic scheme of the air instability focuses on the characterization of the surface contaminate composition, which only

provides ensemble-averaged information of the final products. Using ETEM here, we are able to fill the following knowledge gaps that are required for the passivation design and to solve several puzzles related to the air instability of cathode. Upon revision, we delineate the following key knowledge gaps: morphology; kinetics; and mechanism. As you mentioned that, we use in-situ ETEM clearly reveal that a thin disordered rock-salt layer formed on the NMC811 surface can effectively mitigate the surface degradation by suppressing chemical delithiation, this observation may open a new venue for the development of novel high-energy and high-stability cathodes.

Following your suggestions, we have revised the manuscript to clarify the significance of the in-situ ETEM observation. As citing the introduction part:

“Interfacial reactions that leads to surface passivation may offer an effective means for solving the air instability issues of cathode because it can lower the rate of interfacial reactions, and more critically, turn the detrimental interfacial reactions into beneficial ones.

Unfortunately, the essential elements for establishing the mechanistic understanding of the cathode-air passivation are largely missing. The prevalent scheme of cathode air instability is established based on the techniques such as the X-ray photoelectron spectroscopy (XPS), Thermogravimetric analysis (TGA), and Infrared spectroscopy (IR) that provide ensemble-averaged information, which paint a clear picture of surface contamination composition that is consisted of Li_2CO_3 or $\text{Li}_2\text{CO}_3/\text{LiOH}$ mixture via capturing a snapshot of the reaction products. However, such post-mortem characterizations fail to capture the kinetic evolution of the local interfaces required for defining passivation. Further, the cathode-air interfacial reactions are complicated by the composition of air, consisting of multiple components that likely lead to several concurrent reaction events. The lack of one-to-one correspondence between each single reaction and the overall reaction products masks the nature of the interfacial reactions and fails to screen the species that can effectively slow the side reactions, which leads to some simple yet vague mechanisms behind the interfacial reactions, for example, it is believed that the major event occurring at the interface is a direct reaction between the CO_2 and the surface residue or the cathodes. In general, beside the composition, establishing the passivation principles for the cathodes requires more characteristic information: morphology of the resultant products such as integrity, uniformity, and crystallographic state; the kinetic information related to the initial stage of interfacial reactions to capture the effects of the reaction products on the interfacial reaction kinetics. It is likely due to the lack of the critical information of the cathode-air interfacial reactions, the passivation design has not translated into the battery field for the prevention of air instability. ETEM offers the route to bridge the gap of the initial stage of cathode oxidation: the high vacuum chamber allows to isolate a single gas for reactions, which is facile to unambiguously identify the kinetic protection of each reaction; the high spatial and temporal resolution enables monitoring the nucleation and subsequent growth kinetics, which is critical for capturing the morphology of surface layers at atomic scale.”

Question 2: An as-received pristine Ni-rich NMC particle is not even clean on the surface; LiOH and Li₂CO₃ are present as a residual Li. The content and composition of such residual Li species also have a significant effect on the air-instability of NMCs. How were the NMC samples prepared and treated before the in situ TEM experiment? Please provide more information about the sample sources and the initial surface conditions should be characterized more carefully.

Answer to question 2: We appreciate this great comment and we totally agree with the reviewer about this point. It is generally perceived that the NMCs surface can be covered with either LiOH or Li₂CO₃ or a combination of both, and luckily the TEM imaging is capable of observing the distribution of these surface layer on cathode. Our TEM results suggest that the coverage of Li related residue is not continuous on many surfaces, leaving many bared sections, which serves as a perfect model system to allow us to investigate the passivation behavior. The added supplementary Fig.1 demonstrates a typical example of the pristine particle we selected for the TEM studies. As can be seen, the whole primary particle is clean without visible residue on surface (supplementary Fig.1a), which is further revealed by the high resolution TEM image in which both the edge-plane and base-planes of the NMC622 are completely bared prior to the water exposure (Fig.1b).

To clarify this point, we have added a HRTEM image to illustrate the typical surface structure of a pristine NMC particle (Supplementary Fig. 1). At the same time, we added the following sentences:

“It may be argued that due to the air exposure of NMC, the surface may be already covered with a native layer of LiOH or Li₂CO₃, which can impose significant effect on the Li ions diffusion and affects the in-situ ETEM observation of the passivation behavior. However, HRTEM observation of the pristine sample indicates that the native LiOH or Li₂CO₃ is not continuous, showing features of clean surface segment and therefore lending the convenience for in-situ ETEM observation of the passivation behavior. Apparently, if the thickness of Li related compound is below the critical thickness, the delithiation will continues until reaching a critical thickness; if the thickness of Li compound islands is larger than the critical thickness, the Li compound will remain inert.”

Added as the supplementary Fig.1. HRTEM image showing the representative pristine sample used in the ETEM experiments. (a) TEM image of pristine NMC622. (b) Atomic scale view of the pristine NMC622 surface.

Question 3: The stabilization of the cycled electrode is attributed to the disordered rock-salt layer. However, the surface of the multiple-cycled cathode would be already protected by a thick SEI layer, which may not be completely cleaned away by rinsing with the DMC solvent.

Answer to Question 3: Thank you for raising this great point. We agree that the SEI layers cannot be completely removed simply by DMC rinsing. For our in-situ ETEM studies, following the FIB lift out of the sample, the sample is checked very carefully with TEM imaging to select the rocksalt surface without pre-existing CEI layers to carry out the in-situ ETEM studies (as can be seen in Fig.5g).

To clarify this point, we added the following sentences: “For the studies of the surface rocksalt structure on the surface passivation, following the FIB lift out of the sample, the sample is checked very carefully with TEM imaging to select the rocksalt surface without pre-existing CEI layers to carry out the in-situ ETEM studies (Fig.5g).”

REVIEWERS' COMMENTS:

Reviewer #1 (Remarks to the Author):

The authors addressed all of my questions and comments satisfactorily. I recommend its publication in Nature Comms.

Reviewer #2 (Remarks to the Author):

The manuscript is improved after the careful revision. The reviewers concerns are well addressed. I recommend publication of this revised manuscript.

REVIEWERS' COMMENTS:

Reviewer #1 (Remarks to the Author):

The authors addressed all of my questions and comments satisfactorily. **I recommend its publication in Nature Comms.**

Reply: We appreciate the reviewer's suggestion of publication of the work without further modifications.

Reviewer #2 (Remarks to the Author):

The manuscript is improved after the careful revision. The reviewers concerns are well addressed. **I recommend publication of this revised manuscript.**

Reply: We appreciate the reviewer's positive comment and the suggestion of publication of the manuscript as is.